# A Type Ib Crustin from Deep-Sea Shrimp Possesses Antimicrobial and Immunomodulatory Activity

**DOI:** 10.3390/ijms23126444

**Published:** 2022-06-09

**Authors:** Yu-Jian Wang, Li Sun

**Affiliations:** 1CAS and Shandong Province Key Laboratory of Experimental Marine Biology, Institute of Oceanology, Center for Ocean Mega-Science, Chinese Academy of Sciences, Qingdao 266071, China; wangyujian@qdio.ac.cn; 2Laboratory for Marine Biology and Biotechnology, Pilot National Laboratory for Marine Science and Technology, Qingdao 266237, China; 3College of Earth and Planetary Sciences, University of Chinese Academy of Sciences, Beijing 100049, China

**Keywords:** crustin, WAP domain, bactericidal, deep-sea

## Abstract

Crustins are small antimicrobial proteins produced by crustaceans. Of the many reported crustins, very few are from deep sea environments. Crustins are categorized into several types. Recently, the Type I crustin has been further classified into three subtypes, one of which is Type Ib, whose function is unknown. Here, we studied the function of a Type Ib crustin (designated Crus2) identified from a deep-sea crustacean. Crus2 has a whey acidic protein (WAP) domain and a long C-terminal region (named P58). Recombinant Crus2 bound to peptidoglycan (PGN), lipoteichoic acid (LTA), and lipopolysaccharide (LPS), and killed Gram-positive and Gram-negative bacteria by permeabilizing the bacterial cytomembrane. Consistently, Crus2 dramatically attenuated the inflammatory response induced by LPS and LTA. Disruption of the disulfide bonds in the WAP domain abolished the bactericidal ability of Crus2, but had no effect on the bacterial binding ability of Crus2. Deletion of the C-terminal P58 region moderately affected the antimicrobial activity of Crus2 against some bacteria. P58 as a synthesized peptide could bind bacteria and inhibit the bactericidal activity of Crus2. Taken together, these results revealed different roles played by the WAP domain and the P58 region in Type Ib crustin, and provided new insights into the antimicrobial and immunomodulatory functions of crustins.

## 1. Introduction

Antimicrobial peptides (AMPs) are widely distributed in animals. They are small peptides with various antimicrobial activities against diverse microorganisms [1]. AMPs share certain common features, such as overall positive charge and surface amphiphilicity, that allow them to disrupt the target cells by binding to the cytoplasmic membrane of the cells and/or entering the cells [2,3,4]. With the increase of bacterial antibiotic resistance that has become a global health problem, the search for alternative antimicrobials is an urgent need [5,6]. Compared to antibiotics, AMPs have relatively low resistance development potential. Hence, AMPs have been considered good candidates as antibiotic substitutes [6,7].

Crustins are the largest and a very diverse AMP family in the classified crustacean AMPs [8]. They are usually hydrophobic, cysteine-rich cationic polypeptides containing a whey acid protein (WAP) domain that forms a four-disulfide bond structure [9,10]. Traditionally, crustins were categorized into three main types [11]. Type I crustins exhibit typical crustin characteristics, with four conserved cysteine residues after the signal peptide followed by a C-terminal WAP domain [12,13]. Type II crustins contain two regions rich in glycine and cysteine, respectively, which are replaced by a proline and/or arginine-rich region in Type III crustins [14,15]. All types of crustins possess antibacterial activities. Type II and III crustins can target both Gram-positive and Gram-negative bacteria, while Type I crustins mainly target Gram-positive bacteria [16]. Recent research has expanded the classification of crustins. The Type I crustins are divided into three subtypes, i.e., Ia, Ib, and Ic [17]. Type Ia is the previous Type I; Type Ib has a long random sequence at the C-terminus; Type Ic contains two cysteine-rich regions [17]. The biological functions of the new subtypes, i.e., Ib, and Ic, remain to be investigated.

Deep-sea hydrothermal vents are extreme environments that nurture thriving ecosystems constituted mainly by microorganisms and macro–invertebrates such as shrimp [18]. In particular, the Alvinocarididae family is the dominant shrimp in deep-sea vents, and has been found to encode a variety of antibacterial molecules [19,20,21]. In a previous study, we studied the antibacterial activity of a Type Ia crustin, Crus1, from Alvinocarididae in the deep sea of Manus Basin [22]. In the present work, we characterized another crustin, Crus2, from *Rimicaris* sp., which differs in structure from Crus1. We investigated the antibacterial activity of Crus2 and its dependence on the different domains, in particular the unique C-terminal region characteristic of Type Ib crustins. In addition, we also examined the immunomodulatory activity of Crus2. Our results add new knowledge to the immune function and working mechanism of deep sea crustins.

## 2. Results

### 2.1. Crus2 Is a Type 1b Crustin

Crus2 has 164 amino acid residues, a calculated molecular mass of 18.04 kDa, and a predicted pI of 8.42. Crus2 is predicted to possess an N-terminal signal peptide, a WAP domain with a typical ‘four-disulfide core’ structure, and a C-terminal region of 58 residues (named P58) (Figure 1A). Crus2 shares conserved cysteine residues in the WAP domain and the N-terminus with Type I crustins (Figure 1A). It shows the highest sequence identity (48.60%) with a crustin of *Pacifastacus leniusculus* (PI- crustin 2) and shares 40.19% identity with Crus1, the previously reported crustin of *Rimicaris* sp. [22]. Phylogenetic analysis showed that, different from Crus1, Crus2 was clustered together with the Type Ib crustins (Figure 1B). Structural modeling showed that Crus2 mainly contained random coils (Figure 1C). A similar coil-dominant structure was displayed by the Crus2 truncate (named Crus2DC) that lacks the C-terminal P58 region (Figure 1C). Compared to Crus2, Crus2DC exhibited decreased pI (7.8) and a different distribution of surface electrostatic potential (Figure 1D).

### 2.2. Crus2 Binds and Kills Gram-Positive and Gram-Negative Bacteria in a Manner That Is Affected in Degree by the C-Terminal P58 Region

Recombinant Crus2 and Crus2DC were prepared from *E. coli* (Appendix A), with a yield of 6.2 mg and 5.6 mg per liter of culture, respectively. Both proteins exhibited apparent inhibitory and killing activities against bacteria from different sources, including deep sea (Table 1). In general, the MIC (minimal inhibitory concentration) values of Crus2DC were higher than that of Crus2, except for *Micrococcus luteus*. Similarly, the MBC (minimal bactericidal concentration) values of Crus2DC were also higher than that of Crus2. Both Crus2 and Crus2DC showed apparent binding to various bacteria (Figure 2A). Consistently, Crus2 and Crus2DC bound well to peptidoglycan (PGN), lipoteichoic acid (LTA) and lipopolysaccharide (LPS) (Figure 2B). When Crus2 and Crus2DC were pre-incubated with PGN or LTA, the bactericidal effects of the proteins on the Gram-positive *Bacillus cereus* were significantly decreased; similarly, pre-incubation with LPS significantly inhibited the bactericidal activity of Crus2/Crus2DC toward the Gram-negative *Vibrio harveyi* (Figure 2C,D).

### 2.3. Crus2 Kills Bacteria by Inducing Damage in the Bacteria Membrane

Propidium iodide (PI) staining showed that following treatment with Crus2/Crus2DC, strong fluorescence was observed in *B. cereus* and *V. harveyi*, reflecting disruption of the bacterial membrane (Figure 3A). Cytoplasmic membrane depolarization analysis with DiSC3 (5), a membrane potential-sensitive probe, showed that Crus2 and Crus2DC markedly increased the fluorescent signal in DiSC3 (5)-treated bacteria, indicating changes in the membrane potential (Figure 3B). SEM and TEM revealed that the cells treated with Crus2 exhibited severe structure destruction and release of cellular contents (Figure 3C,D). In contrast, Crus2DC-treated bacteria retained relatively a normal morphology and displayed no apparent structural damage.

### 2.4. The Bactericidal Activity of Crus2 Depends on the Conserved WAP Domain and Is Affected by P58

To explore the essentialness of the conserved disulfide bonds of the Type Ib crustin, the recombinant protein of a Crus2 mutant, Crus2M, was prepared, which bears serine substitutions at C91 and C102. Compared to Crus2, the mutant exhibited no significant difference in the binding to bacteria, LPS, or PGN (Figure 4A,B). However, Crus2M had no apparent effect on the survival of *B. cereus* or *V. harveyi* (Figure 4C). To evaluate the functional importance of the C-terminal P58 region of Crus2, P58 was synthesized as a peptide and examined for its anti-bacterial effect. P58 exhibited no killing effect on *B. cereus* or *V. harveyi* (Figure 5A), and caused no apparent disruption of bacterial structure (Figure 5C). However, P58 was able to bind to bacteria and bacterial components (Figure 5B). Furthermore, pre-treatment of the bacteria with P58 significantly increased the ability of the bacteria to resist subsequent Crus2 killing (Figure 5D).

### 2.5. Crus2 Blocks LPS- and LTA-Induced Inflammatory Response

Since Crus2 and P58 were able to bind LPS and LTA, we examined their potentials to suppress LPS/LTA-induced inflammation. The results showed that pre-incubation with Crus2 significantly impaired the ability of LPS and LTA to induce the release of IL-6, IL-1β and TNF-α from J774.1 cells (Figure 6A–C). Pre-incubation with P58 significantly decreased the release of IL-1β and IL-6, but not TNF-α, caused by LPS (Figure 6D–F). In contrast, pre-incubation with P58 significantly increased LTA-induced releases of IL-1β, IL-6, and TNF-α (Figure 6D–F).

## 3. Discussion

Crustins are important innate immune molecules that are widely present in crustaceans [23]. There are accumulating reports on the crustins from coastal or freshwater crustaceans, but studies on crustins from deep-sea environments are limited. Of the documented deep-sea crustins, a type II crustin was extracted from the hydrothermal vent shrimp *Rimicaris exoculata*, and two other crustins were obtained from the hydrothermal vent shrimp *Alvinocaris longirostris* [24,25]. Our previous study identified a Crus1 from *Rimicaris* sp. and classified it as a type I crustin [22]. In the present study, we analyzed another *Rimicaris* sp. crustin, Crus2. Different from Crus1, Crus2 has a relatively long C-terminal region following the WAP domain, and was therefore classified as a member of the newly identified Type Ib crustin. Interestingly, the eight Cys residues in the WAP domain of Crus2 match that of Type Ia crustins and form four conserved disulfide bonds, suggesting that the extra C-terminal P58 region does not affect the conserved structure of WAP.

Crustins are known to be antimicrobial peptides targeting a broad spectrum of microorganisms. Some reports showed that type I crustins exerted antimicrobial effects only on the bacteria of a Gram-positive nature [22,26,27], while type II crustins are effective antimicrobials against Gram-positive as well as Gram-negative bacteria [28,29]. In this study, we found that unlike the reported classical type I crustins, Crus2 inhibited or killed both Gram-positive and Gram-negative bacteria. In agreement with this observation, Crus2 bound well to the surface molecules of both Gram-positive and Gram-negative bacteria. Furthermore, pre-binding to PGN, LTA, or LPS significantly reduced the bactericidal effect of Crus2, suggesting that these cell wall components are involved in the interaction between Crus2 and its target bacteria.

The ability of AMPs to kill bacteria by breaking down the bacterial cytomembrane has been well demonstrated [30,31]. For some crustins, their antibacterial effects also involve destabilization of the bacterial membrane [13,22]. Similarly, we found that treatment with Crus2 rendered the target bacterial cells permeable by PI and DiSC3 (5), indicating decreased membrane integrity and perturbed membrane potential caused by Crus2. Electron microscopy revealed damaged bacterial structures and leakage of the cytoplasmic content. Together these results indicated that like most AMPs, Crus2 killed the target bacteria by inducing membrane disruption. 

The WAP domain of crustin is a key functional structure required for the antimicrobial response, protease inhibition, and potential immunomodulatory activity of crustins [9,32]. A previous study showed that a synthetic WAP peptide displayed bactericidal activity [24]. Another report demonstrated that crustin lost its bactericidal activity when the disulfide bonds were broken or the conserved cysteine in the WAP domain was mutated [22]. In the present study, we found that the C91S-C102S mutant of Crus2 had little bactericidal activity but retained the bacteria binding capacity. These results indicated that, like other types of crustins, the Type Ib crustin also required the disulfide bonds in the WAP domain for effective bacteria killing. 

Proteins with entirely disordered sequences are called intrinsically disordered proteins (IDPs) [33]. Most eukaryotic proteins possess structured and disordered regions, and both are required for the proper functioning of the proteins [33,34]. The Type Ib crustins are a recently established new subtype. They all feature a more than 30 aa C-terminal region that adopts a disordered structure similar to that of IDPs [17]. The biological activity of Type Ib crustin has not been reported, and, in particular, the function of the long C-terminal region is poorly understood. In our study, we found that the C-terminal P58 region in the form of a synthesized peptide bound well to bacteria and bacterial cell wall components. Moreover, the peptide P58 significantly reduced the bactericidal effect of Crus2, suggesting that the peptide probably competed with Crus2 for bacterial binding. Considering that Crus2 exhibited stronger bactericidal activity than Crus2DC, it was possible that in Crus2, the C-terminal P58 region functioned to enhance the binding between the protein and the target bacteria, thereby maximizing the bactericidal activity of Crus2.

In addition to their well-known antimicrobial activities, some AMPs are involved in the regulation of inflammatory responses [35]. For example, a *Mozambique tilapia* AMP (GRN-4) could enhance the innate immune response by up-regulating the expression of inflammatory cytokines [36]. EPI, an orange-spotted grouper AMP, modulated *S. aureus* LTA-induced inflammatory response [37]. However, to our knowledge, no reports on the immune regulation of crustins have been documented. In our study, we found that Crus2 significantly reduced the inflammatory response elicited by LPS and LTA, suggesting that crustins may not only kill invading microorganisms such as bactericides, but also participate in host immune regulation by neutralizing pathogenic factors. 

In conclusion, our study showed that a deep-sea Type Ib crustin, Crus2, killed diverse bacteria, and this killing activity required the stable disulfide bond in the WAP domain. The Type Ib-typical C-terminal region was not essential to the antimicrobial activity of Crus2; however, via its ability to facilitate bacterial interaction, the C-terminal region was required for the optimal bactericidal effect. Furthermore, Crus2 could act as a modulator of inflammatory response by neutralizing LPS and LTA. These results add new insights into the immune function and mechanism of Type Ib crustins.

## 4. Materials and Methods

### 4.1. Bacterial Strains and Culture Conditions

The bacteria used in this study are listed in Table 1 and have been reported previously [20,22]. Briefly, *Streptococcus iniae* was cultured in tryptic soy broth (TSB) (Hopobio, Qingdao, China) at 28 °C; the deep-sea bacteria, i.e., *Bacillus subtilis* G7, *Bacillus cereus* H2, *Bacillus wiedmannii* SR52, *Bacillus cereus* MB1, and *Marinobacter algicola*, were grown at 28 °C in marine 2216E medium. Luria-Bertani broth (LB) medium was used for the growth of all other bacterial strains, and the growth was carried out at 37 °C (for *Escherichia coli*, *Micrococcus luteus*, and *Staphylococcus aureus*) or 28 °C (for *Edwardsiella tarda*, *Vibrio harveyi*, *Vibrio anguillarum*, and *Pseudomonas fluorescens*). For the bacteriostatic assay, *S. iniae* was cultured in TSB medium, while all other bacteria were cultured in Mueller-Hinton broth (MHB) (Hopobio, Qingdao, China).

### 4.2. Bioinformatics and Structural Characterization

Crus2 sequence (GenBank accession number MW448474) was analyzed with SignalP and InterPro to predict signal peptide and conserved domains, respectively. A sequence similarity analysis was performed using BLAST (NCBI). Sequence alignment and output were created with ClustalX 2.0 (SFI, Dublin, Ireland) and DNAMAN 6.0 (Lynnon Biosoft, San Ramon, CA, USA), respectively. The phylogenetic analysis was performed with the neighbor-joining (NJ) method using MEGA 6.0 (Mega Limited, Auckland, New Zealand), and the reliability was assessed by 1000 bootstraps. The structure models of Crus2 and Crus2DC (a Crus2 truncate that lacks the C-terminal P58 region) were constructed using the I-TASSER online server, and structural images were generated using the PyMOL Molecular Graphics System (Schrödinger, New York, NY, USA) (ver. 3.7).

### 4.3. Preparation of Recombinant Proteins

The coding sequence of Crus2 without signal peptide was synthesized by BGI Technology (Beijing, China). The Crus2DC sequence was obtained by PCR with the primer pair F1/R1 (Appendix A) using Crus2 as the template. Crus2 and Crus2DC were ligated into pET28a (Sangon Biotech, Shanghai, China) at the Nde I/Xho I sites using ClonExpress II One Step Cloning Kit (Vazyme, Nanjing, China), yielding pETCrus2 and pETCrus2DC. *E. coli* Transetta (DE3) (TransGen Biotech, Beijing, China) was transformed with the plasmids. After growing in LB medium at 37 °C to OD_600_ 0.6, the transformants were induced to express the recombinant proteins by adding isopropyl-beta-d-thiogalactoside (IPTG) (0.06 mM) into the culture. The bacterial growth was shifted to 16 °C and continued for 12 h. The cells were then collected by centrifugation and disrupted by sonication. Recombinant proteins were purified with Ni-NTA affinity columns (QIAGEN, Germantown, TN, USA) as reported previously [21,22]. Sodium dodecyl sulfate-polyacrylamide gel electrophoresis (SDS-PAGE) was applied to examine the proteins. A BCA Protein Assay Kit (Beyotime, Shanghai, China) was used to determine protein concentration. To prepare the Crus2 mutant bearing C91S and C102S substitution, reverse PCR was performed with the primers F2/ R2 (Appendix A) using pETCrus2 as the template, resulting in the plasmid pETCrus2-C91S. The second PCR was performed with the primers F3/ R3 (Appendix A) and pETCrus2-C91S as the template, resulting in the plasmid pETCrus2-C91SC102S. The purification of the Crus2 mutant protein was performed as described above for the purification of Crus2.

### 4.4. Peptide Synthesis

The peptide P58 was synthesized by Sangon Biotech Co., Ltd. (Shanghai, China). The peptide was 5’-FITC-tagged and purified to >95% using high performance liquid chromatography. The peptides were dissolved in PBS (pH 7.4) prior to use.

### 4.5. Antibacterial Assay

Determination of the minimal inhibitory concentration (MIC) and minimal bactericidal concentration (MBC) was performed as reported previously [38]. Briefly, bacteria were grown to OD_600_ 0.5–0.6. The bacterial cells were resuspended in MHB to 10^5^–10^6^ CFU/mL in a 96-well microtiter plate. A two-fold dilution series (2–64 μM) of Crus2 or Crus2DC were each placed into the plate. The plate was incubated at 37 °C or 28 °C for 20–24 h, and the culture in the plate was diluted serially and plated on MH agar plates. The MIC was defined as the lowest concentration of Crus2 or Crus2DC that caused no visible bacterial growth to occur. For MBC, it was defined as the concentration of Crus2 or Crus2DC that killed 99.9% of the tested bacteria within an incubation period of 20–24 h. To examine the effect of bacterial components on the bactericidal activity of Crus2 and Crus2DC, Crus2 and Crus2DC were each pre-incubated with 200 μg/mL of PGN, LTA, or LPS (Sigma, St Louis, MO, USA) for 1 h and then incubated with bacteria for 2 h. The colonies of bacteria on the plate was counted as above. To evaluate the inhibitory effect of P58 on the bactericidal ability of Crus2, bacteria were first pretreated with 32 μM P58 or PBS (control) for 1 h at 28 °C, and then treated with 1 × MBC of Crus2 for 2 h at 28 °C. The growth of the bacteria was determined as described above.

### 4.6. Binding of Crus2 and Crus2DC to Bacteria 

Protein binding to the bacteria and bacterial components was determined by ELISA as described previously [21]. Briefly, bacteria (10^8^ CFU/mL) or bacterial cell wall components (PGN, LTA, or LPS, 200 μg/mL) were each added to a 96-well microtiter plate. The plate was incubated at 4 °C for 12–16 h, followed by washing in PBST (PBS with 0.05% Tween 20) three times. Two hundred microliters of 5% skim milk (Solarbio, Beijing, China) dissolved in PBST was added to the plate. After incubation at 37 °C for 2 h, the plate was washed as above. Two hundred microliters of Crus2, Crus2DC, or PBS (control) was added to the plate. After incubation at 37 °C for 1 h, the plate was washed as above. The bound protein was detected by adding anti-His antibody conjugated with HRP (1/1000 dilution) (ABclonal, Hubei, China) into the plate. After incubation at 37 °C for 1 h, the plate was washed five times with PBST. The TMB substrate solution (Tiangen, Beijing, China) was added to the plate for color development. ELISA stop solution was then added to the plate. The absorbance at 450 nm was measured using a multifunctional microplate reader. The binding index = OD_450_ of protein/OD_450_ of PBS.

### 4.7. PI staining Assay and Electron Microscopy

*B. cereus* MB1 and *V. harveyi* were cultured in corresponding media to OD_600_ of 0.8. The bacteria were resuspended in PBS to 1 × 10^6^ CFU/mL. Crus2 or Crus2DC (1 × MBC) was added to the bacteria, and the mixture was incubated at 28 °C for 2 h. The cells were stained with a PI Staining Kit (BestBio, Shanghai, China) and examined using a fluorescence microscope (TiS/L100, Nikon, Tokyo, Japan). For electron microscopy, the above treated bacterial samples were processed as previously described [22] and observed with scanning electron microscopy (S-3400N, Hitachi, Tokyo, Japan) and transmission microscopy (HT7700, Hitachi, Tokyo, Japan).

### 4.8. Membrane Depolarization Assay

The cytoplasmic membrane depolarization induced by Crus2/Crus2DC was measured as reported previously [39]. Briefly, *B. cereus* MB1 and *V. harveyi* were cultured in corresponding media to OD_600_ 0.6 and placed in a HEPES buffer (5 mM HEPES with 20 mM glucose and 100 mM KCI, pH 7.4) to OD_600_ 0.05. The bacteria were then treated with DiSC3(5) (Macklin, Shanghai, China) (0.4 µM). After incubation in the dark for 30 min, the mixture was quenched at 22–25 °C. The samples were then placed into black 96-well microtiter plates. Crus2 (2 × MBC), Crus2DC (2 × MBC), valinomycin, or PBS (control) was added to the plates. Fluorescence change was recorded using a multifunctional microplate reader (620 nm and 670 nm for excitation and emission, respectively).

### 4.9. Pro-inflammatory Cytokine Detection

LPS and LTA (5 μg/mL) were each incubated with Crus2 or P58 (32 μM) or PBS (control) for 1 h at room temperature. J774.1 cells were cultured overnight in a 24-well plate (5 × 10^5^ cells/well) and incubated with Crus2/P58-treated LPS/LTA or untreated LPS/LTA for 24 h at 37 °C. The culture supernatant was collected and used to measure IL-1β (EMC001b, Neobioscience, Shenzhen, China), IL-6 (SEKM0007, Solarbio, Beijing, China) and TNF-α (SEKM0034, Solarbio, Beijing, China) by ELISA according to the manufacturer’s instructions.

### 4.10. Statistical Analysis

Statistical analyses were carried out with GraphPad Prism 7 (GraphPad Software Inc, San Diego, CA, USA) and SPSS 17.0 software (SPSS Inc., Chicago, IL, USA). For two groups, statistical significance was performed with a Student’s *t* test. All data are presented as mean ± SD. *p* < 0.05 was considered statistically significant.

## Figures and Tables

**Figure 1 ijms-23-06444-f001:**
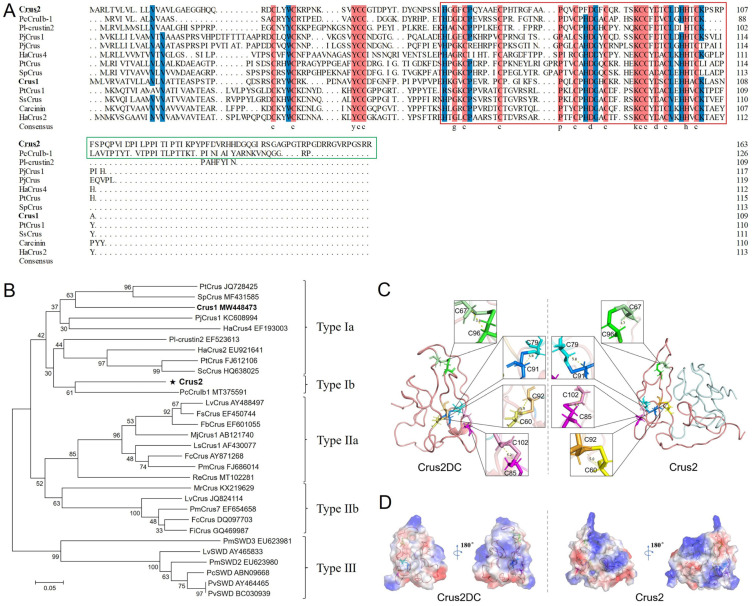
Crus2 sequence and structure. (**A**) The Crus2 sequence was aligned with that of Type I crustins. Dots indicate gaps. Red color indicates consensus residues. Blue color indicates ≥75% identity. The WAP domain and the C-terminal region are boxed with red and green lines, respectively. (**B**) Phylogenetic analysis of Crus2. The numbers indicate bootstrap values based on 1000 replications. The asterisk represents the crustin of this study. In (**A**,**B**), the crustin names are followed by their respective GenBank accession numbers. (**C**) The predicted structures of Crus2 and Crus2DC. The disulfide bonds are shown in yellow (C60–C92), green (C67–C96), blue (C79–C91), and pink (C85–C102). (**D**) The surface electrostatic potential of Crus2 and Crus2DC.

**Figure 2 ijms-23-06444-f002:**
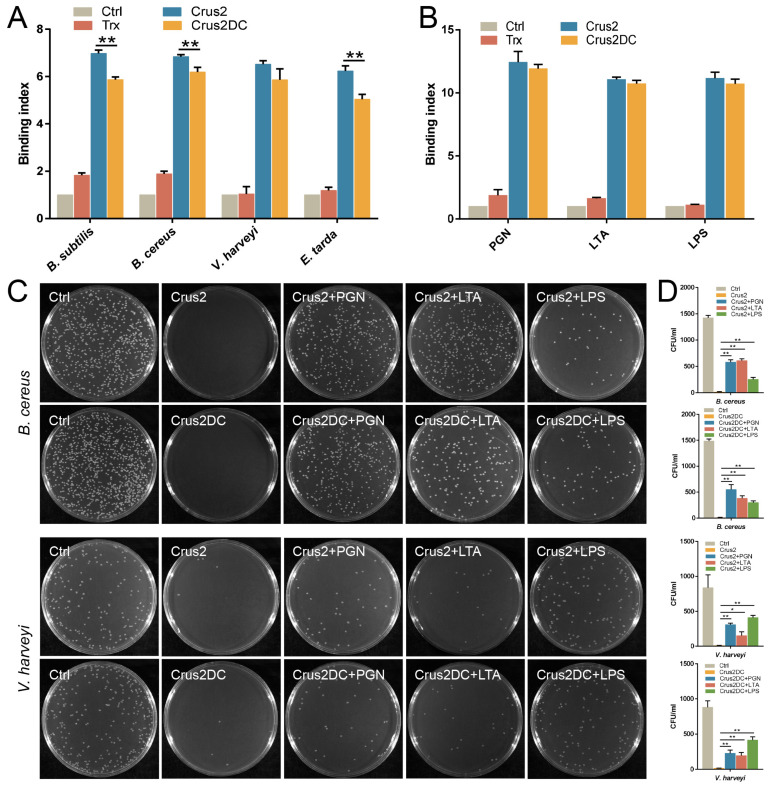
Binding of Crus2 and Crus2DC to bacteria and bacterial components. (**A**) Bacteria were incubated with or without (control, Ctrl) Crus2, Crus2DC, or rTrx, and the binding was detected by ELISA. (**B**) Peptidoglycan (PGN), lipoteichoic acid (LTA), and lipopolysaccharide (LPS) were incubated with or without (Ctrl) Crus2, Crus2DC, or rTrx, and the binding was detected as above. (**C**,**D**) *Bacillus cereus* and *Vibrio harveyi* were incubated with Crus2 or Crus2DC, or Crus2/Crus2DC that had been pre-treated with PGN, LTA, or LPS. The control bacteria were incubated with PBS. After 1 h of incubation, the bacteria were plated on MH agar plates and observed after 20–24 h incubation (**C**). The numbers of colony-forming units (CFU) on the plates were counted (**D**). Values are shown as means ± SD (*n* = 3). *n*, the number of replicates. **, *p* < 0.01; *, *p* < 0.05 (Student’s *t* test).

**Figure 3 ijms-23-06444-f003:**
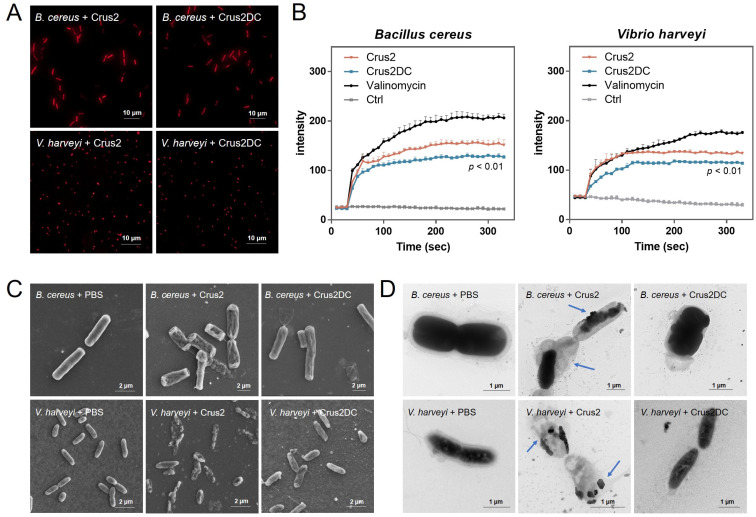
The effect of Crus2/Crus2DC on bacterial structure. (**A**) Bacteria (*Bacillus cereus* or *Vibrio harveyi*) were treated with Crus2 or Crus2DC. After treatment, PI was used to stain the cells, and the stained cells were observed with a fluorescence microscope. (**B**) The above bacteria were first treated with DiSC3 (5) and then treated with or without (Ctrl) Crus2, Crus2DC, or valinomycin, and the fluorescence of the cells was subsequently determined. (**C**,**D**) *B. cereus* and *V. harveyi* were incubated with Crus2, Crus2DC, or PBS. After incubation, scanning (**C**) or transmission (**D**) electron microscopy was applied for structural observation. The blue arrows indicate the breakdown of cellular structure and the release of cellular contents.

**Figure 4 ijms-23-06444-f004:**
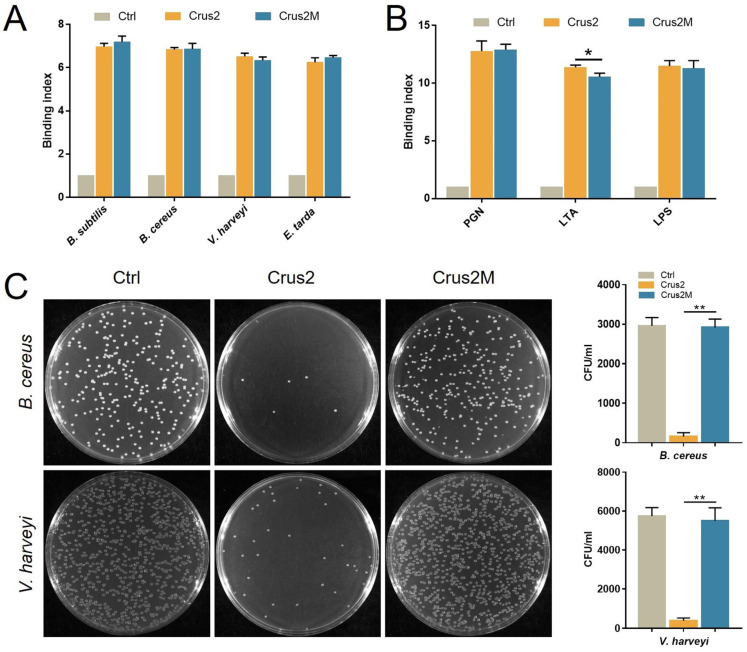
The bacterial binding and bactericidal activities of Crus2M. (**A**) *Bacillus subtilis*, *Bacillus cereus*, *Vibrio harveyi* and *Edwardsiella tarda* were incubated with or without (Ctrl) Crus2 or Crus2M. The binding was subsequently detected by ELISA. (**B**) Peptidoglycan (PGN), lipoteichoic acid (LTA), and lipopolysaccharide (LPS) were incubated with or without (Ctrl) Crus2 or Crus2M, and the binding was detected as above. (**C**) *B. cereus* and *V. harveyi* were treated with or without (Ctrl) 32 μM Crus2 or Crus2M. After treatment, bacterial growth on MH agar plates was observed. The colony-forming units (CFU) on the plates are shown in the bar graphs. Data are the means triplicate experiments and shown as ±SD. ** and * represent *p* < 0.01 and *p* < 0.05, respectively (Student’s *t* test).

**Figure 5 ijms-23-06444-f005:**
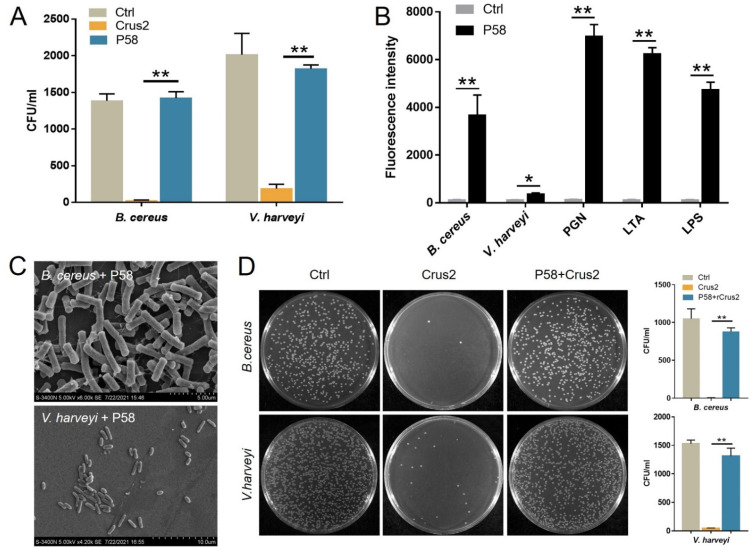
The bacterial binding and bactericidal activities of P58. (**A**) *Bacillus cereus* and *Vibrio harveyi* were treated with or without (Ctrl) Crus2 or P58 for 1 h. After treatment, bacterial growth on MH agar plates was determined. CFU, colony-forming units. (**B**) Bacteria, peptidoglycan (PGN), lipoteichoic acid (LTA), and lipopolysaccharide (LPS) were each incubated with or without (Ctrl) FITC-labeled P58, and the binding was determined by measuring fluorescence intensity. (**C**) *B. cereus* and *V. harveyi* were incubated with P58 for 2 h. After incubation, scanning electron microscopy was applied for structural observation. (**D**) Crus2 was incubated with *B. cereus* and *V. harveyi* that had been pre-treated with or without (Ctrl) P58. After incubation, bacterial growth on MH agar plates was determined. The CFU numbers are shown in the graphs. Data are the means triplicate experiments and shown as ± SD. ** and * represent *p* < 0.01 and *p* < 0.05, respectively (Student’s *t* test).

**Figure 6 ijms-23-06444-f006:**
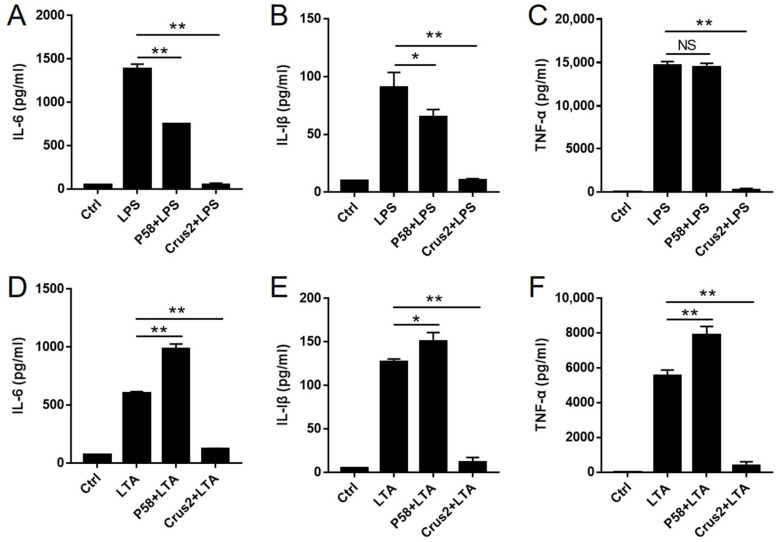
The effects of Crus2 and P58 on LPS- and LTA-induced inflammatory response. (**A**–**C**) J774.1 cells were incubated with or without (Ctrl) lipopolysaccharide (LPS) that had been pre-incubated with or without Crus2 or P58 for 24 h. The release of inflammatory cytokines (IL-6, IL-1β, and TNF-α) was then detected. (**D**–**F**) J774.1 cells were incubated with or without (Ctrl) lipoteichoic acid (LTA) that had been pre-incubated with or without Crus2 or P58 for 24 h. The release of inflammatory cytokines (as above) was then detected by ELISA. Data are the means triplicate experiments and shown as ± SD. ** and * represent *p* < 0.01 and *p* < 0.05, respectively (Student’s *t* test).

**Table 1 ijms-23-06444-t001:** The MIC and MBC of Crus2 and Crus2DC against different bacteria.

Bacteria	MIC (μm)		MBC (μm)	
	Crus2	Crus2DC	Crus2	Crus2DC
Gram-positive bacteria				
*Bacillus subtilis* G7	16	32	32	64
*Bacillus cereus* MB1	16	32	32	64
*Bacillus cereus* H2	32	16	32	32
*Bacillus wiedmannii* SR52	16	32	16	32
*Micrococcus luteus*	16	16	32	16
*Staphylococcus aureus*	16	16	32	32
*Streptococcus iniae*	32	32	32	32
Gram-negative bacteria				
*Vibrio harveyi*	16	32	16	32
*Edwardsiella tarda* TX1	32	32	32	32
*Vibrio anguillarum*	32	32	—	—
*Escherichia coli*	—	—	—	—
*Pseudomonas fluorescens*	—	—	—	—
*Marinobacter algicola*	—	—	—	—

—: No activity was detected.

## Data Availability

All data are included in the article or its Appendix A.

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
