# Peer review of "A Type Ib Crustin from Deep-Sea Shrimp Possesses Antimicrobial and Immunomodulatory Activity"

_ijms, 2022, doi:10.3390/ijms23126444_

Round 1

Reviewer 1 Report

The article by Wang et al. reports the characterization of a type Ib crust from the shrimp Rimicaris sp.

I find the article very nice and well structured. Furthermore, I think this class of proteins is very interesting and useful for applications in the field of AMPs. This protein can be defined as an IDP, being more or less completely unstructured, even though IDPs are usually acidic proteins. In any case, I think the role of the C-terminal tail is very interesting and represents an important difference between type I crustins, which is reflected in a broad spectrum activity against Gram-positive and Gram-negative bacteria, unlike type Ia. In the future, it would be worth investigating the shorter C-terminus of the cruIb-1 crustin of Penaeus chinensis type Ib.

I believe the article is suitable to be printed in this form

Minor points

Adding information on the yield of the homogeneous protein

Add something more in the Introduction or some other section about the complete absence of structure, which seems to be a common feature among the various crustins so that they can be possibly defined as IDPs

IPTG concentration so low?

Trasetta (DE3) : to be corrected

Reviewer 2 Report

The manuscript "A Type Ib crustin from deep-sea shrimp possesses antimicrobial and immunomodulatory activity” is dedicated to the identification of antimicrobial peptide Type Ib crustin (Crus2) from the shrimp Rimicaris sp. isolated from an extreme habitat. This study examined the antimicrobial activity of Type Ib crustin against gram-positive and gram-negative bacteria. The relationship the bactericidal activity of Crus2 and the conserved WAP domain and P58 was studied. The effect of peptides Crus2 and Crus2DC on bacterial morphology was demonstrated with SEM and TEM. The role of the C-terminal P58 sequence was assessed both for bactericidal activity and ability to bind with peptidoglycan, lipoteichoic acid and lipopolysaccharide.

There are some minor corrections:

1. The text contains numerous grammatically incorrect or clumsy sentences, e.g:

Abstract: clumsy sentence «Disruption of the disulfide bonds in the WAP domain abolished the bactericidal ability, but not the bacterial binding capacity, of Crus2»

Abstract: “degree of activity” sounds rather inappropriate in the case of antimicrobial properties

2. Line 102 – italic font needed

3. Table 1 in the text is mentioned before Figure 2, but it is located on the page after Figure 2. Rearrangement would make this section easier to comprehend.

4. Panels with CFU counts on Figure 2 should have their own letter (D) and a legend separate from panel C.

5. On figures 3C and 3D, the species name V.harveyi should not be capitalized. The same correction needed on Figure 5C for B. cereus.
